# The Effects of GCSF Primary Prophylaxis on Survival Outcomes and Toxicity in Patients with Advanced Non-Small Cell Lung Cancer on First-Line Chemoimmunotherapy: A Sub-Analysis of the Spinnaker Study

**DOI:** 10.3390/ijms24021746

**Published:** 2023-01-16

**Authors:** Shobana Anpalakhan, Prerana Huddar, Roya Behrouzi, Alessio Signori, Judith Cave, Charles Comins, Alessio Cortellini, Alfredo Addeo, Carles Escriu, Hayley McKenzie, Gloria Barone, Lisa Murray, Gagan Bhatnagar, David J. Pinato, Christian Ottensmeier, Fabio Gomes, Giuseppe Luigi Banna

**Affiliations:** 1Portsmouth Hospitals University NHS Trust, Portsmouth PO6 3LY, UK; 2The Christie NHS Foundation Trust, Manchester M20 4BX, UK; 3Section of Biostatistics, Department of Health Sciences University of Genoa, 16126 Genoa, Italy; 4Department of Medical Oncology, University Hospital Southampton NHS Foundation Trust, Southampton SO17 1BJ, UK; 5Bristol Haematology and Oncology Centre, Bristol Royal Infirmary, Bristol BS2 8HW, UK; 6Department of Surgery and Cancer, Imperial College London, Hammersmith Hospital Campus, London SW7 2AZ, UK; 7Medical Oncology, Fondazione Policlinico Universitario Campus Bio-Medico, Via Alvaro del Portillo, 00128 Roma, Italy; 8Oncology Department, HUG—Hopitaux Universitaires de Geneve, 1205 Geneva, Switzerland; 9The Clatterbridge Cancer Centre NHS Foundation Trust, Liverpool L7 8YA, UK; 10University Hospitals of Northamptonshire, Northampton NN1 5BD, UK; 11Division of Oncology, Department of Translational Medicine, University of Piemonte Orientale, 28100 Novara, Italy; 12The Clatterbridge Cancer Centre NHS Foundation Trust, University of Liverpool, Liverpool L7 8YA, UK

**Keywords:** NSCLC, lung cancer, immunotherapy, GCSF, neutropenia, prophylaxis, immune-related toxicity, neutrophil-to-lymphocyte ratio (NLR), neuthrophils, outcome, overall survival

## Abstract

GCSF prophylaxis is recommended in patients on chemotherapy with a >20% risk of febrile neutropenia and is to be considered if there is an intermediate risk of 10–20%. GCSF has been suggested as a possible adjunct to immunotherapy due to increased peripheral neutrophil recruitment and PD-L1 expression on neutrophils with GCSF use and greater tumour volume decrease with higher tumour GCSF expression. However, its potential to increase neutrophil counts and, thus, NLR values, could subsequently confer poorer prognoses on patients with advanced NSCLC. This analysis follows on from the retrospective multicentre observational cohort Spinnaker study on advanced NSCLC patients. The primary endpoints were OS and PFS. The secondary endpoints were the frequency and severity of AEs and irAEs. Patient information, including GCSF use and NLR values, was collected. A secondary comparison with matched follow-up duration was also undertaken. Three hundred and eight patients were included. Median OS was 13.4 months in patients given GCSF and 12.6 months in those not (*p* = 0.948). Median PFS was 7.3 months in patients given GCSF and 8.4 months in those not (*p* = 0.369). A total of 56% of patients receiving GCSF had Grade 1–2 AEs compared to 35% who did not receive GCSF (*p* = 0.004). Following an assessment with matched follow-up, 41% of patients given GCSF experienced Grade 1–2 irAEs compared to 23% of those not given GCSF (*p* = 0.023). GCSF prophylaxis use did not significantly affect overall or progression-free survival. Patients given GCSF prophylaxis were more likely to experience Grade 1–2 adverse effects and Grade 1–2 immunotherapy-related adverse effects.

## 1. Introduction

Granulocyte-colony-stimulating factor (GCSF) prophylaxis is used in patients on chemotherapy based on their risk of developing febrile neutropenia. This risk is determined by factors such as elderly age and neutrophil count. GCSF prophylaxis should be implemented in chemotherapy regimens with a high risk of febrile neutropenia (>20%) and at least considered if there is an intermediate risk (10–20%) [1,2]. During the COVID-19 pandemic, GCSF prophylaxis was used more widely to reduce the risk of adverse outcomes associated with COVID-19 infection [3,4]. There are, however, important safety issues to consider when GCSF is used, including a surge in pulmonary inflammation, macrophage activation and an increase in the neutrophil-to-lymphocyte ratio (NLR), which could subsequently lead to respiratory deterioration in COVID-19-positive patients [5,6].

There have been reports of GCSF increasing peripheral neutrophil recruitment and programmed death-ligand 1 (PD-L1) expression on neutrophils, as well as a greater decrease in tumour volume in cancers with higher GCSF expression. These mechanisms could result in increased efficacy of immune checkpoint inhibitors (ICIs); therefore, GCSF has been put forward as a potential adjuvant therapy to ICIs [7].

The administration of GCSF can result in both neutrophilia and a reduction in lymphocyte count, thus increasing the NLR [8] and potentially influencing the associated prognosis. The NLR has been shown to be a useful prognostic indicator in patients with non-small-cell lung cancer (NSCLC) as well as a predictor of response to immunotherapy [9,10,11]. Moreover, in patients with renal cell cancers, variations in the NLR have been shown to influence overall survival (OS) and progression-free survival (PFS) [12].

The Spinnaker retrospective study evaluated efficacy outcomes for patients with advanced NSCLC on first-line chemoimmunotherapy and in doing so, developed the prognostic score ‘NHS-Lung score’ to allow for risk stratification of these patients regardless of PD-L1 status [13]. This subsequent analysis following on from the Spinnaker study aimed to explore if the use of GCSF could be clinically beneficial in treatment-naïve patients with advanced NSCLC on chemoimmunotherapy either by a synergistic effect with the immunotherapy resulting in prolonged OS and PFS or by reducing the haematologic toxicity and immunosuppression of the chemotherapy. A differential clinical effect of GCSF use according to the baseline NLR level was also investigated with the hypothesis that different outcomes could be observed following the GCSF stimulation in those patients with higher levels of neutrophils to the detriment of lymphocytes.

## 2. Results

The Spinnaker study included 308 patients of ECOG-PS 0-1 from 7 different centres (16). The characteristics of this patient cohort according to the use of GCSF primary prophylaxis are described in Table 1. As demonstrated, it was noted that patients who received GCSF had a shorter median follow-up duration (*p* < 0.001). The median follow-up duration among those given GCSF prophylaxis was 12.8 months (95% CI: 12.3–13.3 months), while it was 20.7 months (95% CI: 19.0–22.5 months) among those not given any GCSF. The subsequent matched analysis revealed that this difference was not significant (*p* = 0.533). Among the factors that did not differ significantly between patients who received GCSF and those who did not were age, tumour histology, PS, PD-L1 status, pre-treatment steroid use, pre-treatment NLR, pre-treatment SII and the number of metastatic sites. This was confirmed following analysis with matched follow-up. The initial analysis with unmatched follow-up revealed a significant difference in the chemotherapy regimen given to patients on GCSF compared to those not on GCSF (*p* = 0.007). However, when this was repeated with matched follow-up, there was no significant difference found (*p* = 0.867).

Patients who received GCSF primary prophylaxis were more likely to experience Grade 1–2 AEs (*p* = 0.004), and this was confirmed following analysis with matched follow-up. A total of 56% percent of patients receiving GCSF had Grade 1–2 AEs compared to 35% who did not receive GCSF. There was no significant difference in the proportion of patients who experienced Grade 3–4 AEs (*p* = 0.537). A total of 24% percent of patients on GCSF and 19% of patients not on GCSF experienced Grade 3–4 AEs. There were no significant differences in the proportion of patients experiencing irAEs of any grade in the unmatched follow-up analysis. Repeat assessment with matched follow-up in fact revealed a significant difference in Grade 1–2 irAEs, which were more frequent in patients on GCSF (*p* = 0.023). A total of 41% percent of patients on GCSF experienced Grade 1–2 irAEs compared to 23% of patients not on GCSF.

As demonstrated in Figure 1, the use of GCSF primary prophylaxis had no significant impact on median OS. This was 13.4 months (95% CI: 10.4–16.5 months) in patients given GCSF and 12.6 months (95% CI: 9.7–15.4 months) in patients not given GCSF (*p* = 0.948). It also yielded no significant difference in median PFS between those given GCSF with a median PFS of 7.3 months (95% CI: 4.6–10.0 months) and those not given GCSF with a median PFS of 8.4 months (95% CI: 7.5–9.3 months) (*p* = 0.369).

Median OS and PFS among patients with a high NLR did not differ significantly whether GCSF was used or not (*p* = 0.954 and *p* = 0.358, respectively). In patients with a high NLR, the median OS was 13.4 months (95% CI: 8.0–18.9 months) in those given GCSF and 11.8 months (95% CI: 9.1–14.5 months) in those not given GCSF. Median PFS in this same sub-cohort with a high NLR was 5.5 months (95% CI: 4.0–7.0 months) when given GCSF and 6.9 months (95% CI: 5.0–8.8 months) when not given GCSF prophylaxis. Patients with a low NLR also showed no significant difference in median OS or PFS (*p* = 0.924 and *p* = 0.883, respectively) with or without the use of GCSF. Median OS was 13.5 months (95% CI: 11.1–15.9 months) in patients given GCSF and 16.0 months (95% CI: 12.4–19.7 months) in patients not given GCSF. Median PFS was 8.0 months (95% CI: 4.4–11.5 months) for patients on GCSF and 9.0 months (95% CI: 7.3–10.7 months) in patients not given GCSF (Appendix A). In the following analysis with matched follow-up, the differences in OS and PFS among patients with high (*p* = 0.357 and *p* = 0.832, respectively) and low (*p* = 0.932 and *p* = 0.417, respectively) NLR values remained non-significant. 

## 3. Discussion

The results of this analysis have shown that the use of GCSF prophylaxis does not affect the survival outcomes of patients with advanced NSCLC treated with chemoimmunotherapy. It has shown, following analysis with matched follow-up, that patients given GCSF prophylaxis were more likely to experience Grade 1–2 AEs and Grade1–2 irAEs. The key limitations of this analysis include its retrospective nature, the lack of experimental verification and detailed information about the type of GCSF used and timing of GCSF administration, the dynamics and characterisation of the peripheral and tumour-infiltrating immune cells, and NLR changes following the GCSF stimulation and initial unmatched follow-up among the two groups of patients, which introduced a potential time bias, although this was accounted for with a subsequent analysis with matched follow-up. Furthermore, a specific effect of GCSF with immunotherapy could be masked by concomitant chemotherapy. On the other hand, the multicentre and real-life nature of this study allows for the generalisability of its results.

Contrarily to other reports which investigated the use of primary GCSF prophylaxis with chemotherapy regimens characterised by a high risk of febrile neutropenia [14,15], the chemotherapy regimen used in the present series does not have a high risk of febrile neutropenia and, therefore, does not necessarily require GSCF prophylaxis [1,2]. Moreover, this analysis has shown that a multitude of factors used in various prognostic tools, such as age, histology, PS, PD-L1 status, pre-treatment steroids, pre-treatment NLR, pre-treatment SII and the number of metastatic sites, did not predict whether patients received GCSF prophylaxis. The Spinnaker study established the NHS-Lung score, uses the number of metastatic sites, tumour histology and the SII score to guide prognostication [13]. It concluded that more metastatic sites, squamous tumour histology and a higher SII score conferred a poorer prognosis. The Lung Cancer Prognostic Index (LCPI) uses a number of factors, including tumour stage and histology, mutation status, PS, weight loss, smoking history, respiratory comorbidity, sex and age, to determine prognosis [16]. Moreover, elderly age was also deemed an important factor in determining the risk of febrile neutropenia [1]. The Lung Immuno-oncology Prognostic Score-3 (LIPS-3) attributes a poor prognosis to patients with a PS of ≥2, requiring pre-treatment steroids (indicating possible brain or liver metastases or nutritional issues), and an NLR ≥ 4 [17]. PD-L1 expression has also been shown to be a marker for poor prognosis [18,19]. In addition to this, GCSF prophylaxis has been suggested as a potential adjunct to immunotherapy [7]. Despite these reports, these numerous factors were shown not to influence GCSF primary prophylaxis use in this analysis.

Median follow-up was seen to be significantly shorter in the group receiving GCSF. This observation was potentially due to a larger proportion of patients receiving GCSF and commencing follow-up from the start of the COVID-19 pandemic in the United Kingdom in March 2020 compared to other patients who had been followed up since April 2019. Therefore, primary prophylaxis with GCSF was likely adopted as a general precautionary strategy offered during the COVID-19 pandemic.

The administration of GCSF prophylaxis was shown not to affect OS or PFS in this series of patients treated with first-line chemoimmunotherapy for advanced NSCLC. This is in contrast to another study on patients with metastatic pancreatic cancer on FOLFIRINOX chemotherapy that found primary GCSF prophylaxis actually improved OS [20]. The global, randomised ECHELON-1 study on patients with Hodgkin’s lymphoma on brentuximab vedotin and doxorubicin as well as vinblastine and dacarbazine also found an improvement in survival outcomes with the use of GCSF through reduced frequency and severity of AEs, treatment delays and episodes of chemotherapy discontinuations [21]. These discrepancies could be attributed to the toxicity variations with different chemotherapy regimens.

The observations in survival outcomes noted in this analysis were consistent when stratified by NLR values as well. Previous reports had suggested the use of GCSF prophylaxis potentially affects the NLR [11] and, thus, prognosis. Further work, including experimental models, needs to be undertaken to assess the dynamics and characterisation of the peripheral and tumour-infiltrating immune cells and changes in the NLR following GCSF administration and the nature and duration of these changes, taking into account that patients tend to be on relatively short courses of GCSF prophylaxis of 5–7 days. While GCSF has been reported to increase PD-L1 expression on neutrophils and contribute to increased tumour shrinkage in cancers with GCSF expression, further research on the effects of GCSF prophylaxis on these and the duration of these effects is required [7].

There were significantly more patients on GCSF experiencing general Grade 1–2 AEs than those not on GCSF. A potential explanation for this is that side effects attributed to treatment regimens may be in fact secondary to GCSF administration, including myalgia, fatigue and skin rash. The matched follow-up analysis revealed a significant difference in Grade 1–2 irAEs as well. This may demonstrate a true increase in mild immunotherapy-related side effects with GCSF use. As GCSF use has been linked to potentially increasing the NLR, this may explain the higher incidence of Grade 1–2 irAEs among patients given GCSF prophylaxis in this analysis. The REISAMIC registry prospective study found that an elevated NLR was associated with more severe immunotherapy-related toxicities [22]. On the other hand, this finding could also be explained by the similar side effects of chemotherapy and immunotherapy or even treatment overlap, but these are being labelled as immunotherapy-related effects.

Nonetheless, there were no significant differences in treatment discontinuation rates between the groups despite more Grade 1–2 AEs in patients receiving GCSF prophylaxis. There were also no significant differences in the proportion of patients experiencing Grade 3–4 AEs of any nature. This could also be explained by the only temporarily increased neutrophil counts following GCSF administration. Pre-treatment with steroids may have contributed to patients not experiencing irAEs, although only approximately 10–11% of patients in each group had been on steroids.

## 4. Materials and Methods

The Spinnaker study was a retrospective multicentre observational cohort study focusing on real-world patients with histologically confirmed advanced NSCLC, any PD-L1 tumour proportion score (TPS), no actionable genomic alterations and an Eastern Cooperative Oncology Group Performance Status (ECOG PS) ≤ 1, who had been treated with first-line chemotherapy and pembrolizumab and whose outcome results have recently been reported [13]. Patients were treated in six United Kingdom centres and one Swiss centre between March 2018 and April 2021. Among the collected patient information, the use of primary prophylaxis with GCSF was recorded and defined by the use of any non-pegylated drugs for at least five consecutive days (i.e., filgrastim 300–480 mcg or lenograstim 263 mcg subcutaneous daily), or pegylated ones for one day (i.e., pegfilgrastim 6 mg or lipegfilgrastim 6 mg), following each chemoimmunotherapy cycle starting from the first one. The NLR was calculated as the ratio between neutrophils and lymphocytes from the peripheral blood count of a standard blood test performed within 14 days of the treatment start date. A high NLR was considered a value ≥4 according to the literature-reported cut-off [17]. The systemic immune-inflammatory index (SII) was calculated as the NLR multiplied by the platelet count with the previously reported cut-off value of ≥ 1440 [13].

The primary endpoint of this analysis was to describe the patients’ characteristics and survival outcomes (i.e., OS and PFS) according to the use of GCSF primary prophylaxis. Secondary endpoints included the frequency and severity of adverse events (AEs) and immune-related adverse events (irAEs) as adjudicated by the clinician and graded according to the common toxicity criteria for adverse effects (CTC-AE) version 5.0, and the interaction between GCSF primary prophylaxis and NLR as per the above cut-off in OS and PFS and with PD-L1 status (TPS <1%—negative vs. ≥1%—positive) and chemotherapy regimen in OS.

Clinical data were analysed by descriptive statistics, using percentages for the binary variables and medians for the continuous variables, reporting their respective dispersion values. For the comparison of binary variables, the chi-square test with an acceptable significance value of *p* < 0.05 was performed. The OS was calculated from the treatment start date until death or the date of the last follow-up; the PFS was taken from the treatment start date to disease progression or death from any cause. Patients who had not had any events at the time of the analysis were censored. OS and PFS were estimated using the Kaplan–Meier method and reported as medians with confidence limits (95% CIs), and compared using a two-sided log-rank test with an acceptable significance value of *p* < 0.05 [23]. An interaction test between the GCSF primary prophylaxis and the above factors was performed. The statistical analysis was carried out using the SigmaPlot software version 12.5 (Systat Software, San Jose, CA, USA).

Given the significant difference found in the median follow-up time between patients treated with GCSF primary prophylaxis versus those who did not receive it in the study population, a secondary comparison of the study endpoints was performed using a cohort of patients who did not receive GCSF matched according to their follow-up time. The follow-up time was matched by manually searching within the non-GSCF cohort for a treatment start date by which the median follow-up time coincided with the GCSF cohort; this corresponded to the end of November 2019.

## 5. Conclusions

This analysis has shown that the use of GCSF prophylaxis does not affect OS or PFS in patients with advanced NSCLC on chemoimmunotherapy. There was a higher incidence of Grade 1–2 general and immunotherapy-related side effects in patients given GCSF. Future prospective research is required to further assess these outcomes.

## Figures and Tables

**Figure 1 ijms-24-01746-f001:**
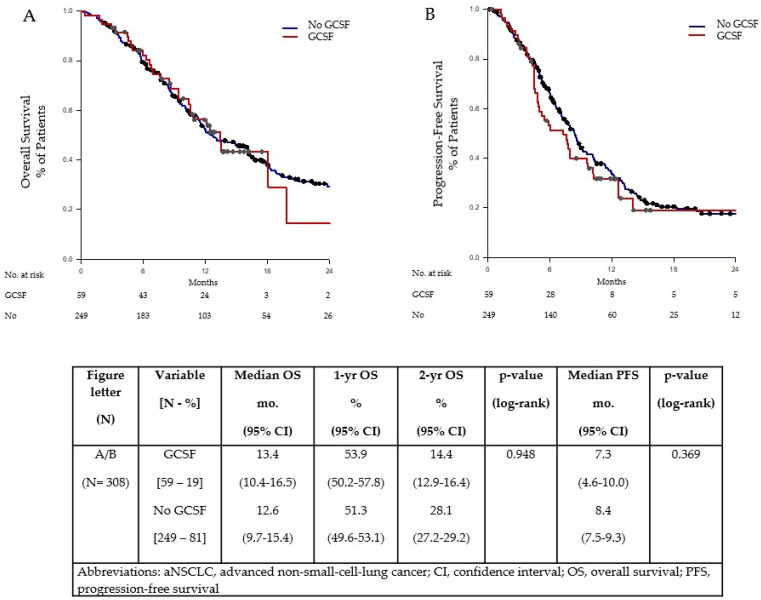
OS (**A**) and PFS (**B**) by GCSF use in patients with aNSCLC treated with chemoimmunotherapy.

**Table 1 ijms-24-01746-t001:** Association of G-CSF use with patient characteristics and outcomes.

Characteristic	G-CSF (No. 59) No. (%) [Range]	No G-CSF Unmatched (No. 249) No. (%) [Range]	χ^2^ Test (Log-Rank) *p*-Value	No G-CSF Matched ^a^ (No. 114) No. (%) [Range]	χ^2^ Test (Log-Rank) *p*-Value
Age			0.591		0.919
≥70 years	21 (36)	77 (31)		40 (35)	
<70 years	38 (64)	172 (69)		74 (65)	
Gender			0.714		0.463
Male	31 (53)	140 (56)		68 (60)	
Female	28 (47)	109 (44)		46 (40)	
Smoking history			0.494		0.951
Never	3 (5)	22 (9)		7 (6)	
Former/Current	56 (95)	227 (91)		107 (94)	
Histology			0.550		0.643
Squamous	12 (21)	39 (16)		27 (25)	
Adenocarcinoma	46 (79)	200 (84)		80 (75)	
Other	1 (2)	10 (4)		7 (6)	
ECOG PS			0.351		0.161
0	28 (47)	99 (40)		39 (35)	
1	31 (53)	150 (60)		72 (65)	
Stage			0.275		0.654
IIIB/IVA	22 (37)	115 (46)		48 (42)	
IVB	37 (63)	134 (54)		66 (58)	
BMI			0.306		0.713
Underweight/Normal	27 (46)	135 (54)		57 (50)	
Overweight/Obese	32 (54)	114 (46)		57 (50)	
Number of metastatic sites			0.706		0.935
<3	41 (69)	164 (66)		80 (70)	
≥3	18 (31)	85 (34)		34 (30)	
Brain metastases	6 (19)	25 (10)	0.833	13 (11)	0.992
Liver metastases	4 (7)	33 (13)	0.249	14 (12)	0.389
PD-L1 IHC Ab ^b^			0.495		0.416
Negative	34 (61)	131 (55)		58 (53)	
Positive	22 (39)	109 (45)		52 (47)	
NA	3 (5)	9 (4)		4 (4)	
Pre-treatment steroids	6 (10)	27 (11)	0.933	13 (11)	0.992
Pre-treatment NLR ≥ 4	34 (58)	130 (52)	0.545	59 (52)	0.566
Pre-treatment SII ≥ 1440	30 (51)	124 (50)	1.000	58 (51)	0.875
Type of chemotherapy			**0.007**		0.867
Cisplatin–Pemetrexed	3 (5)	21 (8)		4 (4)	
Carboplatin–Pemetrexed	45 (76)	195 (78)		87 (76)	
Carboplatin–Paclitaxel	11 (19)	33 (13)		23 (20)	
Best response ^c^			0.744		0.706
CR/PR	37 (65)	161 (68)		68 (64)	
SD	12 (21)	40 (17)		19 (18)	
PD	8 (14)	37 (16)		20 (19)	
NA	1 (2)	11 (4)			
G1/2 AE–G1/2 irAE	33 (56)–24 (41)	86 (35)–76 (31)	**0.004**–0.199	29 (25)–26 (23)	**<0.001–0.023**
G3/4 AE–G3/4 irAE	14 (24)–9 (15)	48 (19)–40 (16)	0.537–0.993	18 (16)–17 (15)	0.285–0.869
Treatment discontinuation	14 (24)	58 (23)	0.920	19 ()17)	0.359
Follow-up, median, mo. [95%]	12.8 [12.3–13.3]	20.7 [19.0–22.5]	**(<0.001)**	13.9 [9.9–17.8]	0.533
Deaths	29 (49)	151 (61)	0.143	56 (49)	0.875
OS, median, mo. [95% CI]	13.4 [10.4–16.5]	12.6 [9.7–15.4]	(0.948)	11.3 [9.0–13.5]	0.357
1 yr OS [95% CI]	53.9 [50.2–57.8]	51.3 [49.6–53.1]		44.7 [42.3–47.3]	
2 yr OS [95% CI]	14.4 [12.9–16.4]	28.1 [27.2–29.2]		18.2 [16.1–20.9]	
PFS, median, mo. [95% CI]	7.3 [4.6–10.0]	8.4 [7.5–9.3]	(0.369)	7.5 [5.8–9.2]	0.832
COVID-19-positive	2 (3)	8 (3)	0.734	3 (3)	0.844
COVID-19 deaths					
Death rate	1 (3)	7 (5)	0.836	2 (4)	0.555
Positive rate	1 (50)	7 (88)	0.843	1 (33)	0.576

Abbreviations: Ab, antibody; AE, adverse events; BMI, body mass index; CI, confidence interval; CR, complete response; ECOG PS, Eastern Cooperative Oncology Group Performance Status; G, grade; IHC, immunohistochemistry; mo., months; No. number; NA, not assessable; NLR, neutrophil-to-lymphocyte ratio; OS, overall survival; PD, progressive disease; PD-L1, programmed cell death-ligand-1; PFS, progression-free survival; PR, partial response; SD; stable disease; SII, systemic immune-inflammatory index; yr, year. ^a^ Matched by follow-up time (cut-off date for non-GCSF cohort 25.11.19). ^b^ Negative, TPS > 1%; positive, TPS 1−49%; high, TPS ≥ 50%. ^c^ By RECIST version 1.1 criteria. Statistically significant values in bold.

## Data Availability

The datasets generated and analysed during the current study are not publicly available as they are part of the confidential medical record but are available from the corresponding author on reasonable request.

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
