# Peer review of "The Effects of GCSF Primary Prophylaxis on Survival Outcomes and Toxicity in Patients with Advanced Non-Small Cell Lung Cancer on First-Line Chemoimmunotherapy: A Sub-Analysis of the Spinnaker Study"

_ijms, 2023, doi:10.3390/ijms24021746_

Round 1
Reviewer 1 Report
The reviewed article aimed to assess the effects of granulocyte colony-stimulating factor on overall survival (OS) and progression-free survival (PFS) in NSCLC patients after chemoimmunotherapy.
The study was performed between 2018-2021 as a multicentre observational cohort Spinnaker study where six UK centers and one Swiss were included. Finally, 308 patients were included. The authors provided detailed data about patient characteristics and outcomes. Discussed results and added the key limitations of the study. I do not have major comments on this analysis. As a minor, authors should format literature according to journal requirements.
Author Response
I do not have major comments on this analysis. As a minor, authors should format literature according to journal requirements.
Re: thank you very much for appreciating our manuscript. We have now formatted the references according to the journal requirements.
Reviewer 2 Report
In this manuscript, the authors investigated the Effects of GCSF Primary Prophylaxis on Survival. The primary endpoints were OS and PFS. The secondary endpoints were the frequency and severity of AEs and irAEs. They concluded that GCSF prophylaxis use did not significantly affect overall or progression-free survival. Patients given GCSF prophylaxis were more likely to experience Grade 1-2 adverse effects and Grade 1-2 immunotherapy-related adverse effects.
Questions about this manuscript:
1. The figures are of low quality and some errors in the pictures e.g. the red lines ”aNSCLC” still in the pictures.
2. The authors only do a retrospective analysis, and no further experiment verification.
3. The background description is not enough to clearly state the questions and hypothesis.
4. Some grammar errors need to be carefully checked.
Author Response
Questions about this manuscript:
1. The figures are of low quality and some errors in the pictures e.g. the red lines ”aNSCLC” still in the pictures.
Re: thank you. We have improved the quality of the figures and corrected the errors.
2. The authors only do a retrospective analysis, and no further experiment verification.
Re: thank you. We appreciate and agree with your viewpoint about the lack of prospective or experimental verification, particularly regarding the specific effect of GCSF with immunotherapy only, the dynamics and characterisation of the peripheral and tumour-infiltrating immune cells and NLR changes following the GCSF stimulation. For this reason, we had already acknowledged the study limitations immediately in the Discussion, which we have now enriched with the above concepts and a spur to provide experimental models with further research.
3. The background description is not enough to clearly state the questions and hypothesis.
Re: thank you. We have enriched the background with further evidence and clarified the clinical question of the analysis and hypothesis.
4. Some grammar errors need to be carefully checked.
Re: thank you. The manuscript has now been carefully reviewed and amended where needed.
Reviewer 3 Report
A retrospective study was carried out by Shobana et al. to comprehend the effects of GCSF primary prophylaxis in advanced non-small cell lung cancer patients who had first-line chemoimmunotherapy.
I believe that the majority should change this study. Blows are important criticism for writers.
First, the approaches did not include information on how to match the follow-up period of individuals who did not get GCSF. The classification of matched and unmatched groups leads to a scenario that is completely unintelligible.
Second, there is not much uniqueness in this research. The major findings of this publication have been discussed in a number of earlier works, including PMID: 12966417. In the background section, the author should provide a thorough explanation of the goal and significance of this study. To reflect the benefits and advancement of current research, it is also vital to compare the published conclusions with the new findings. I find it difficult to understand the study's new significance in the current manuscript. This section needs to be changed.
Third, there is just one queue in use. Exists any bias here? Can the author incorporate further queue data, such as PMID: 10931450, for integrated analysis?
Author Response
I believe that the majority should change this study. Blows are important criticism for writers.
Re: thank you. We hope that the revised version of this manuscript based on your and other reviewers’ comments could be satisfactory.
First, the approaches did not include information on how to match the follow-up period of individuals who did not get GCSF. The classification of matched and unmatched groups leads to a scenario that is completely unintelligible.
Re: thank you. We provided that information following what was already stated in the Materials and Methods paragraph as follows: “Given the significant difference found in the median follow-up time between patients treated with GCSF primary prophylaxis versus those who did not receive it in the study population, a secondary comparison of the study endpoints was performed using a cohort of patients who did not receive GCSF matched by their follow-up time. The follow-up time was matched by manually searching within the non-GSCF cohort a treatment start date by which the median follow-up time coincided with the GCFS cohort; it corresponded to the end of November 2019.”
As stated in the material and methods, the analysis was planned for the GCSF and the unmatched cohort, while that with the matched one was post-hoc and relative to the significant difference found in the median follow-up, which was related to the adoption of the GCSF prophylaxis during the Covid-19 pandemic. However, the distribution of patients according to their characteristics and results, summarised in Table 1 and described in the related manuscript paragraph, were quite consistent when comparing the GCSF with the non-GCSF either unmatched or matched cohorts, with the latter reinforcing the conclusions of neither beneficial nor detrimental effect of GCSF on overall or progression-free survival but a higher rate of grade 1-2 overall and immunotherapy-related adverse effects in patients given GCSF prophylaxis.
Second, there is not much uniqueness in this research. The major findings of this publication have been discussed in a number of earlier works, including PMID: 12966417. In the background section, the author should provide a thorough explanation of the goal and significance of this study. To reflect the benefits and advancement of current research, it is also vital to compare the published conclusions with the new findings. I find it difficult to understand the study's new significance in the current manuscript. This section needs to be changed.
Re: thank you. We appreciate your viewpoint and apologise for not being clear enough with the study background. Indeed, this point was also raised by another reviewer. Our study's novelty relies on the clinical effect of prophylactic GCSF with immunotherapy, although given in combination with chemotherapy. Noteworthy, the combined chemotherapy regimen is not expected to give a high risk of febrile neutropenia (>20%) and, therefore, does not necessarily require GSCF prophylaxis. We have enriched the background with further evidence and clarified the clinical question of the analysis and hypothesis. Particularly, we clarified the study question as it follows: “This subsequent analysis following on from the Spinnaker study aims to explore if the use of GCSF could be clinically beneficial in treatment-naïve patients with advanced NSCLC on chemoimmunotherapy either by a synergistic effect with the immunotherapy resulting in prolonged OS and PFS or by reducing the haematologic toxicity and immunosuppression of the chemotherapy. A differential clinical effect of GCSF use according to the baseline NLR level was also investigated with the hypothesis that different outcomes could be observed following the GCSF stimulation in those patients with higher levels of neutrophils to the detriment of lymphocytes.”. Consequently, in the Discussion paragraph we acknowledged among study limitations that: “…a specific effect of GCSF with immunotherapy could be masked by concomitant chemotherapy.” We mentioned the study you quoted in the Discussion.
Third, there is just one queue in use. Exists any bias here? Can the author incorporate further queue data, such as PMID: 10931450, for integrated analysis?
Re: thank you for this comment. We have enriched the study limitation section reported at the beginning of the discussion paragraph as follows: “The key limitations of this analysis include its retrospective nature, the lack of experimental verification and detailed information about the type of GCSF used and timing of GCSF administration, or the dynamics and characterisation of the peripheral and tumour-infiltrating immune cells and NLR changes following the GCSF stimulation, as well as initial unmatched follow up among the two groups of patients which introduced a potential time bias although this is accounted for with a subsequent analysis with matched follow up. Furthermore, a specific effect of GCSF with immunotherapy could be masked by the concomitant chemotherapy.”
Furthermore, we discussed the suggested evidence as it follows: “Contrarily to other reports which investigated the use of primary GCSF prophylaxis with chemotherapy regimens characterised by a high risk of febrile neutropenia (PMID: 12966417, PMID: 10931450), the chemotherapy regimen used in the present series is not expected to give a high risk of febrile neutropenia and, therefore, does not necessarily require GSCF prophylaxis (1,2). Moreover, this analysis has shown that a multitude of factors used in various prognostic tools such as age, histology, PS, PD-L1 status, pre-treatment steroids, pre-treatment NLR, pre-treatment SII and the number of metastatic sites did not predict whether patients received GCSF prophylaxis.”
Round 2
Reviewer 2 Report
1. The authors revised several questions that were raised last time, but the figures are still of a low quality, and I don’t think it is acceptable for publication.
2. All the results provided are negative conclusions which do not show much meaningful things for the clinical practice.
3. The method and the figure legend are not clearly described.
Reviewer 3 Report
The authors responded to most of my concerns. I have no further questions.